# Pulsed Electromagnetic Fields (PEMF)—Physiological Response and Its Potential in Trauma Treatment

**DOI:** 10.3390/ijms241411239

**Published:** 2023-07-08

**Authors:** Jonas Flatscher, Elizabeth Pavez Loriè, Rainer Mittermayr, Paul Meznik, Paul Slezak, Heinz Redl, Cyrill Slezak

**Affiliations:** 1Ludwig Boltzmann Institute for Traumatology, The Research Center in Cooperation with AUVA, 1200 Vienna, Austria; 2AUVA Trauma Center Vienna—Meidling, 1120 Vienna, Austria; 3Department of Physics, Utah Valley University, Orem, UT 84058, USA

**Keywords:** biophysical forces, clinical use, regeneration, cellular signaling, electromagnetic field, PEMF

## Abstract

Environmental biophysical interactions are recognized to play an essential part in the human biological processes associated with trauma recovery. Many studies over several decades have furthered our understanding of the effects that Pulsed Electromagnetic Fields (PEMF) have on the human body, as well as on cellular and biophysical systems. These investigations have been driven by the observed positive clinical effects of this non-invasive treatment on patients, mainly in orthopedics. Unfortunately, the diversity of the various study setups, with regard to physical parameters, molecular and cellular response, and clinical outcomes, has made it difficult to interpret and evaluate commonalities, which could, in turn, lead to finding an underlying mechanistic understanding of this treatment modality. In this review, we give a birds-eye view of the vast landscape of studies that have been published on PEMF, presenting the reader with a scaffolded summary of relevant literature starting from categorical literature reviews down to individual studies for future research studies and clinical use. We also highlight discrepancies within the many diverse study setups to find common reporting parameters that can lead to a better universal understanding of PEMF effects.

## 1. Introduction

It is considered a fact that cells and corresponding tissues are responsive to changes in their environment, such as mechanical stress, fluctuations in pH and O_2_ levels, or fluid flow [1]. External mechanical forces are specifically relevant in wound healing [2], but they also play a central role in bone formation. This particular process was already described in 1892 by Wolf, who indicated that bone growth and remodeling are induced by external forces [3,4]. Interestingly, in the late 1950s, Yasuda et al. revealed that mechanical stresses on the bone give rise to piezoelectrically generated currents [5], which later were connected to the orientation and pattern of collagen and its response to mechanical loads [4]. The piezoelectric constant depends on the angle of pressure direction [5] and the humidity of bone [6] and is roughly a tenth of that of a quartz crystal [5], although newer piezoelectric materials can achieve piezoelectric constants several magnitudes higher [7].

These electrical properties of collagen can be interpreted as a mechanism for osteocytes to sense areas with stress. Indeed, generated currents have been connected to the stimulation of osteoblasts and bone formation, giving an excellent example of how biophysical forces are part of the environmental impact on cells, tissues, and organs. The fact that collagen and similar proteins and structures are found in many different tissues and organs throughout the body makes this type of observed electro-physical interaction universally relevant to the human body [8]. 

Pulsed electromagnetic fields (PEMFs) have been suggested to elicit a similar biological tissue and cellular response by directly inducing electrical currents in the therapy zone while forgoing the aforementioned mechanical agitation. This treatment concept is far from new; in fact, PEMF was introduced as a treatment in the 1970s by Bassett and colleagues [9], and continues to be an interesting clinical treatment strategy with ongoing new indications. Following FDA approval in 1979, this treatment approach has been in clinical use for several decades in treating orthopedic indications, such as bone formation, non-unions, osteoarthritis, and more. In these treatments, a wide range of treatment parameters such as EM pulse shapes and sequences have been considered, and the effectiveness of PEMF therapies has become more elucidated, not only in orthopedics [10,11]. Nevertheless, a clear understanding of the underlying molecular mechanisms and associated robust clinical outcomes remains elusive because of its diverse use. This currently leaves researchers and clinicians to navigate an extensive, yet diverse, portfolio of publications, which can be complicated to homogenize.

Therefore, the aim of this review is to (1) systematically and critically organize resources on the established physical background and cellular response, (2) demonstrate and summarize the most relevant mechanisms of PEMF that have been described so far, and (3) explore the use of this technology in the clinic, particularly in the fields of trauma and regeneration. The examined reviews and summaries are provided in Table 1. Examining the complex web of overlapping references and structural similarities would require a robust statistical dataset for all publications. However, compiling overlapping references using the REST API by Crossref (Lynnfield, Massachusetts) shows clear qualitative differences in the available data, as not every publication does provide further information on DOIs or PMIDs, which were also considered for interlinking. This makes automation and, thus, a systematic statistical consideration almost impossible, without correcting for every publication manually. As AI tools are becoming a reliable researcher support, these tasks may be more feasible in future.

## 2. Technology

PEMF therapy is a non-invasive treatment that applies intermittent, current pulse-generated magnetic field pulses over a short time frame Δt to living tissue, using a pulse repetition frequency  f. An additional electrical current is subsequently induced in conductive materials, leading to a secondary effect of PEMF in tissue. Due to the short pulse duration and the typically low application pulse repetition frequency, the magnetic field is activated only for a fraction of the therapy time. For the generation of the pulses, different waveform shapes are used with PEMF, ranging from rectangular and triangle to sinusoidal, thus including a range of harmonic field frequencies. The therapeutic exposure times may vary from a few minutes up to several hours.

Interestingly, even though PEMF treatment has been available for years, there are hardly any guidelines for categorizing PEMF. One such approach would be a differentiation by frequency. The IEEE categorizes electromagnetic frequencies into ULF (Ultra Low Frequency, <3 Hz), ELF (Extremely Low Frequency, 3 Hz–30 kHz), and VLF (Very Low Frequency, 30 kHz–300 kHz) magnetic fields [50], even though different ranges for each definition can be found [21,51]. A general problem with the frequency definition is that, as the name PEMF suggests, the applied magnetic fields are pulsed, and not continuous waves. As a result, the term frequency is most commonly associated with the pulse repetition frequency, disregarding the field variations within the pulse (referred to as carrier frequency or field frequency). As the signal itself often consists of a continuous train of sine waves, or other signal shapes, which are intermittently turned on and off with the pulse repetition frequency, this field frequency can be several magnitudes higher, often in the kHz region [20], even reaching frequencies in the MHz range [52]. It is, therefore, essential to clearly distinguish between the pulse repetition frequency and the field frequency.

Another term repeatedly found is HI (High Intensity) PEMF [53], with a strong peak magnetic field generated in the process. The strength or amplitude of the magnetic field is commonly reported using the magnetic flux density B. Here, again, we see a wide range, from a few microTesla to several Tesla in HI PEMF, although lower flux densities are the most used. This may be attributable to the comparable simplicity of generating weaker field strengths.

### 2.1. Magnetic Field

This section deals with a closer look at the basic biophysical effects associated with the applied magnetic field. The ability of a material to respond to an external magnetic field is described through its permeability, which, therefore, also describes how the applied magnetic field is influenced if a medium is present. Depending on its magnetic properties, the material aligns with (paramagnetic) or opposes (diamagnetic) the external material field, resulting in being either repelled or attracted [54], thus altering the local magnetic field.

On a macroscopic level, human tissue has a negligible influence on an external magnetic field and doesn’t attenuate it, as biological tissue is only very weakly diamagnetic or paramagnetic [55], depending on which organ is observed. Nevertheless, this does not mean that, on the microscopic level, it cannot influence those tissues’ or organs’ molecular constitutions. For example, inside the human body, a few molecules containing ferromagnetic and paramagnetic metal elements (e.g., iron and manganese) are present, which react more strongly to the presence of an external magnetic field and, thereby, interact with their environment [54]. However, while blood contains a high percentage of iron, due to the presence of hemoglobin, its magnetic properties are also subject to oxygenation levels; for example, venous blood has a stronger paramagnetic property than arterial blood [54].

### 2.2. Electrical Field

An applied time-varying magnetic field also creates a changing electric field, which can impact tissues. If a conductive material is present, the change of the magnetic field density B can induce an electric field E, according to the 3rd Maxwell Equation:∇ × E=δB/δt

Interestingly, the tissues’ interaction with an electrical field is much more frequency-dependent than the magnetic field [55]. The low-frequency electrical fields do not penetrate well into tissue, instead favoring the development of an electric current on the surface [28,50,51]. However, applied penetrating magnetic fields can be used to induce an electrical field in deeper tissue in PEMF therapy [50]. The crucial part of this equation is that the electrical field is only induced during the change of the magnetic field. Therefore, the gradient of the pulse is the important part, not merely the pulse duration Δt, nor the pulse frequency f. In a traditional simple coil design, the maximum gradient can be estimated with:$${\left(\delta B/\delta t\right)}_{max}=V\mu_{0}Nf/L,$$
where V is the recharge voltage, $\mu_{0}$ is the permeability of free space, N is the number of coil turns, f is a geometric factor of the coil, and L is its inductance [56]. The magnetic signal shape also determines the duration of the electrical field. Square waves induce only a short, but high-intensity, electrical field. At the same time, a lower gradient found in a triangle shape generates a weaker electrical field which is present for a longer period. To estimate the electrical field, either the circuit parameters must be known, or the magnetic signal form must be measured—both parameters are seldomly stated in published studies, which prohibits a post-hoc association with differential biophysical responses.

### 2.3. Tissue Interaction

The applied PEMF field therefore influences tissues in two ways: Firstly, the magnetic field creates a force on tissue-resided molecules which depend on their magnetic reactive properties, and secondly, the induced electrical field, which exerts a force on the ions present in the tissue; both result in a forced movement of ions or charged particles, such as proteins [39,57]. Panagopoulus et al. [36] propose that a low field frequency may have more potential to be bioactive than static magnetic fields, and a pulsed magnetic field may be twice as effective as a continuous one. While the magnetic flux density B or the magnetic field strength H is mostly specified in PEMF studies, the actual signal form, which is essential for determining the induced electric field as specified above, is rarely provided. Without it, effectively comparing studies is hampered, as observed effects cannot be associated with direct magnetic effects or those of induced electrical fields, and a study replication may result differently.

In contrast to most other biophysical therapy methods, such as photo biomodulation, extracorporeal shock wave therapy (ESWT), and electrical field therapy, the magnet field can penetrate the human body without much resistance and associated losses—where, in ESWT, the changes in tissue density (e.g., bones and the lung) impede signal propagation, visible light is easily scattered, and, in electrical stimulation, the human skin or bones work effectively as electrical insulating barriers [39].

Even the excitation of nerves can be observed through the induction of an electrical current, which results in muscle contraction, especially when it is close to the so-called motor threshold (MT) [40]. This type of both therapeutical and training interaction is also found in EMT (Electromagnetic Muscle Training). Nevertheless, the application of strong magnetic fields in the proximity of the heart should be handled with special precautions [55].

In an attempt to categorize complex interactions, Mansourian and Shanei [27] conducted a meta-analysis on the effect and parameters of PEMF, based on reviewing 92 publications from 1999–2019. They found that the effect of PEMF differs between cell type (stem cells) and origin (human/animal). Especially, osteosarcoma cells seem to be very sensitive to PEMF stimulation. According to this analysis, pulse repetition frequencies higher than 100 Hz with magnet flux densities between 1 mT and 10 mT lead to the highest presence of a cellular response, although this may vary depending on the cell type and stage of growth [19,20,28,58]. Also, repeated applications over a prolonged period of more than 10 days show a higher effect than shorter periods, while a prolonged acute exposure lasting more than 24 h seems to be less effective than an acute exposure with less than 24 h application time. Surprisingly, triangle waveforms have the highest cellular response (78.46%), while square waves only showed a cellular response in around 40% of the experiments [27]. However, not including the pulse duration makes this information difficult to interpret, as any generated electrical field depends on the gradient of the magnetic field. If the field frequency (spectra) was controlled in all the studies, using triangular pulses would mean that the observed effects depended on the oscillating of a low electrical field, rather than on short, but stronger, pulses, or that the electrical field has less effect than anticipated. Unfortunately, without this information, drawing a conclusion is almost impossible, highlighting the necessity for improved study designs and documentation standards.

### 2.4. Technology Comparison

A lot of different minimal invasive therapy methods are in clinical use, but the sheer amount of abbreviations and different mechanism may be confusing at first. Some of them also overlap in technology or are a more refined definition of another already established therapy method. To give a clear overview of the most prominent therapies and their differences, a short comparison is presented in Table 2.

### 2.5. Study Design

For in vitro studies in cell cultures, the peak magnetic flux density is usually well documented. A Helmholtz, or large solenoid coil, is used to create a uniform magnetic field in the targeted area, which is often measured and documented during the study. Unfortunately, the exact signal shapes, as mentioned before, are missing in most studies. These inconsistencies between studies are often too large to make a viable comparison feasible, making it very hard to associate the direct biophysical response [19].

In vivo applications and clinical studies in animals often use solenoid coils [70], either by aerial application, such as placing several coils below a rat cage, or applying a targeted single coil close to the desired tissue. The direction of the magnetic field may not influence the outcome [56], but these coils have non-linear magnetic field strength distributions outside their cores, and their reach is very limited. The magnetic flux density at the end of a coil can already drop by a factor of two over what is measured at its center, and increasing the coil’s length only further increases the drop-off in the magnetic field at the ends, in exchange for a more uniform distribution inside the coil [71]. An improved design is recommended to increase the penetration depth, such as batwing coils or double cone coils [72], but also to improve the magnetic field range by adding a magnetic core. Using larger Helmholtz coils for in vivo application can result in a more controlled distribution of the magnetic field [56,73,74].

In clinical studies, the magnetic flux densities are seldom measured, which brings up the great disadvantage of only reporting a single magnetic flux density value, as specified by the manufacturer. These values are not representative of the actual in situ applied magnetic field, but of reference values, as measured at the coil’s center, where the field is strongest. For single coils, the induced electric field can be estimated using the Faraday’s law [39], and, for multiple coils or complex coil geometries, the magnetic field has to be measured or simulated. The absence of this information in most studies misses out on a great opportunity to provide a better physical characterization of the therapy, as the tissue barely influences the magnetic field distribution [75], and, in contrast to other non-invasive biophysical stimulation techniques such as ESWT, the measured magnetic field in free space can be readily translated into the in vivo application to characterize its impact.

A major problem of the available publications is the aforementioned missing parameters of the signal shape, which is discussed in Markov et al. [28]. As utilization ranges from square functions to triangle shapes to a block of sine waves, it is rarely possible to compare different studies. As mentioned above, the gradient of the magnetic flux density is responsible for the magnitude and duration of the induced electrical current, and this can vary widely, even for a similar electrical signal form, depending on the properties of the used coils. In summary, the parameters presented in Table 3 would be needed to describe a PEMF signal [76], and should be considered a documentation standard.

This is particularly challenging, as the used devices are either custom-made or field measurements for the multitude of PEMF device manufacturers that are not readily available. In addition, the field frequencies of clinical devices are, on average, significantly higher than those in in vivo or in vitro studies [19]. The replication of most published results is often not possible, due to the absence of an identical setup, as a result of the discrepancies mentioned above, but also due to the lack of a characterization of the electromagnetic field [28]. Therefore, a comparative study would be necessary to link the mentioned parameters with the observed biological effects to improve the clinical application; a review of already published studies may not be sufficient, due to the lack of measurements and settings. Table 4 provides a sample of the diversity of available PEMF devices and the lack of accessible reference data.

### 2.6. Safety and Adverse Effects

Due to the broad range of available PEMF settings, studies covering its whole range are not common. Most publications for electromagnetic fields are in different frequency regimes, such as the power-net 50–60 Hz region [50], Transcranial Magnetic Stimulation (TMS, similar to HI-PEMF, but transcranial), or Magnetic Resonance Imaging [55].

Low frequencies of ELF magnetic fields are non-ionizing and do not increase the temperature noticeably [86], even though electrodes or metal implants could heat up [40]. For strong static magnetic fields, sensory sensations of nausea, vertigo, and a metallic taste are reported, although no evidence of permanent harmful effects could be detected [55]. Biological interactions at low level magnetic fields could be observed [51]. However, the role in carcinogenic effects is still unclear [27], but studies provide evidence suggesting strong magnetic fields as a co-carcinogen with known genotoxins [19]. Therefore, although no direct link to DNA-damaging effects could be found yet [50,75], the “as low as reasonably achievable” (ALARA) recommendation should still be followed. 

Due to the observed synergy between constant and changing magnetic and electric fields (even the magnetic field of the earth at ~30 µT), it is suggested that, even in the low frequency range, the present static field should also be measured [50]. The removal of the static geomagnetic field has been reported to influence cell proliferation, indicating its necessity for a proper biological function [87].

## 3. Cellular and Molecular Response to PEMF

The scope of cellular-based studies on the effects of PEMF-related molecular responses has, so far, led to several reviews on the subject. They all include multifunctional actions of how tissues and organs deal with damage and homeostasis maintenance, which encompass fundamental cellular processes such as apoptosis, proliferation, and differentiation. Here, we critically review these works to link the known biophysical PEMF effects to the many examined PEMF-generated cellular responses and put them in context to the above-mentioned cellular processes.

### 3.1. Direct Cellular Response to PEMF

The direct effect of PEMF on cellular response has been coupled to particular cell membrane channels and adenosine response.

#### 3.1.1. Ion Channels

The cells in our human body are in a constant electro-chemical flux (e.g., K^+^ or Ca^+^ ion gradients), which plays a central role in cell membrane function and is, therefore, equally important in a myriad of cellular activities. It is no surprise, then, that studies have been focusing on the influence of PEMF on these membrane ion channels [88], most specifically, focusing on the effects on calcium ion signaling, where many biological effects are mediated by intracellular Ca^2+^ changes. Here, the release of Ca^2+^ ion and the direct activation of PEMF on voltage-gated calcium channels (VGCCs) is of great relevance. The activation of these membrane-bound channels normally produces downstream effects that affect processes such as cell metabolism, apoptosis, proliferation, and inflammation [89]. It has been shown that PEMF stimulation also leads to similar membrane effects, resulting in a Ca^2+^ influx, which triggers further cellular signals [4]. Studies that have been done on this particular ion channel activation have not been done on similar cells or parameters, which has been pointed out in several reviews, questioning the robustness of these findings. Nevertheless, it is important to mention that there is no decisive evidence that other voltage sensitive channels are not also activated by PEMF [37,90,91]. This necessitates more targeted investigations, with regard to, not only the sensitivity (i.e., PEMF biophysical parameters) of the VGCCs, but also the characterization of similar PEMF-sensitive ion channels in the cell membrane. Importantly, this research might trigger further implementation or regulation of this type of therapy in drug delivery, due to ion channels being one of the six main pharmacological targets in drug discovery [77].

#### 3.1.2. Adenosine and its Receptors

Adenosine has also been directly connected to PEMF response. This endogenous purine nucleoside has various roles in biological processes and is derived from ATP, ADP, and AMP. The intracellular levels of adenosine are usually maintained at a low level; it is when the cell is in higher demand of energy due to different activities (e.g., metabolic activity) that the extracellular levels of adenosine rise. The function of this molecule is mediated by cell membrane receptors A1, A2A, A2B, and A3A. These G protein-coupled receptors, particularly those which elevate intracellular cAMP (i.e., A2A and A2B), can serve as sensors of microenvironmental changes and promote a protective cell/tissue response. They are usually coupled; A1 with A3A, and A2A with A2B [92,93].

These receptors are found to be differentially expressed in human cells. For example, A2A can be found in chondrocytes, synoviocytes, osteoblasts (in combination with A3A), dermal fibroblasts, keratinocytes, neutrophils, neurons, and endothelial cells, while A2B is expressed in keratinocytes and other epithelial cells [1,15,92,93,94,95,96].

PEMF has been shown to induce the cell membrane A2A and A3 expression in multiple cell types, meaning that this stimulation involves only two of the four receptors, which would conclude that its effects specifically target cells and tissues that have these two particular receptors [1,15,91,96,97]. This induction of expression has been coupled to elevated cell proliferation and increased inhibition of terminal differentiation and activation of osteoclasts [96].

### 3.2. Essential Cellular Processes and PEMF

#### 3.2.1. Apoptosis

The onset of reactive oxygen species (ROS) after PEMF treatment has been pointed out in several reviews, and the subsequent impact on cell viability and apoptosis has been elaborated [12,15,22,37,88,90,91,93,96,97,98]. It has been proposed that the accumulation of ROS or oxidative stress may cause the upregulation of heat shock proteins (Hsp70, HIF-1), leading to cell damage. Additionally, PEMF has also been shown to influence c-Jun N-terminal kinase (JNK) signaling and caspase-dependent apoptotic response [12], possibly due to a ROS-mediated response. If ROS is indeed one of the first molecular events elicited by PEMF, then the previously mentioned change in Ca^2+^ can be seen as an important mechanism by which these ROS-dependent events can take place [91]. 

On the other hand, the increase of nitric oxide (NO) after PEMF exposure on osteoblast has been seen as a potential inhibiting effect on apoptosis and improving cell viability [90]. The differences in apoptotic response have been discussed alongside ROS in the above-mentioned reviews, as well as pointing out the differences in treatment used (e.g., variety in flux density ranging from a few µT up to tens of mT), in cell type (e.g., cells from bone, cartilage, heart, and genetically aberrant cell lines, representing different cancer types) and in experimental setup (e.g., cell density, media conditions, time lapse, analytical format). This is perhaps mostly well elucidated in the review of Barati et al. [12], where the authors set out to look for ways to interpret these conflicting data of pro-/anti-apoptotic effects of PEMF in cell malignancy. Interestingly, PEMF does not seem to give rise to a dose response pattern, as PEMF, by itself, does not elicit clear apoptotic behavior or biological effects on malignant cells.

Nevertheless, reports have suggested that even very low pulsed magnetic flux density can give rise to biological effects. This is especially shown in results from studies where chemotherapeutics were used in combination with PEMF, showing an “active” role of PEMF in the therapeutic response. However, it has also been suggested that the PEMF exposure conditions might not be as important in the PEMF response pattern as the biological state of the experiment (cell type, experimental setup). All these variables leave the gate open for many different interpretations, with regards to apoptosis, with no clear main conclusion, and, therefore, the authors suggest a strategy where comparisons between studies are made with clear set criteria from the beginning. By following this approach, Barati et al. concluded that the PEMF elicited apoptotic effects, as seen in the studies on malignant cells, so far could be divided into three groups: (1) PEMF exposure prior to treatment with apoptosis-inducing substances (AIS), (2) PEMF exposure and AIS simultaneously, (3) AIS exposure, followed by PEMF [12]. The first option can lead to the immediate activation of different cell defense mechanisms, triggering the cellular repair systems before the inclusion of AIS, to which the cells will already be in a defense mode with a robust protective system, leading to a lower apoptotic response. The two latter setups result primarily in ROS response, which leads to an increase in apoptosis, instead. Even though there are still many questions with regards to the mechanisms of action for these three scenarios, one may conclude that PEMF is suitable for new combination treatment strategies in combination with chemotherapeutics, where the sequence of treatment and a certain window of physical parameters (mentioned by Barati et al.) are used.

Furthermore, the apoptotic response is not only related to the intracellular response, but also to the intercellular communication, where both microvesicles and miRNAs play a role and can elicit different cellular actions in neighboring cells. Gianfranco et al. [22] summarized the studies related to PEMF effects on this particular form of signaling and epigenetic regulation, and found that PEMF can indeed trigger a miRNA response related to an apoptotic response (i.e., induction of miR494-3p).

#### 3.2.2. Proliferation

It has been shown in many studies that PEMF affects cell proliferation. Recently, Mansurian et al.’s [27] meta-analysis of PEMF studies (1999–2019) identified more than 30 articles related to the proliferative response to PEMF treatment. As one of the most common forms of cell activity, proliferation is important in both healthy and pathological conditions. Most articles investigating PEMF’s proliferative action have been performed on stem cells, followed by effects on different cancer cells and bone and cartilage cells [27].

##### Effects in Osteoblasts and MSCs

Osteoblasts are very important in bone formation, bone matrix synthesis, and mineralization, and, thus, the focus of many studies is trying to better understand their proliferative response after PEMF therapy. The studies show mixed results, leading to either an increase or inhibition of proliferation after PEMF. Here, the outcome of the response has been connected to the maturity level of the osteoblasts, as well as their microenvironment. With regards to molecular pathways triggered by PEMF in osteoblasts, both calcium channels and adenosine receptors are affected, and a group of molecular pathways (e.g., BMP2, Wnt, mTOR, MAPK) are triggered and lead to a proliferative response and effects on bone formation. Additionally, PEMF also influences osteoclast proliferation, although the response varies in different studies and the molecular response differs from that of osteoblasts, pointing to a RANKL- and Nf-κb-driven response. The review by Zhang et al. also describes the response patterns formed with regard to the pulse frequency and the intensity of the treatment for osteoblasts and osteoclasts, and, additionally, shows the difference between electromagnetic parameters [48]. In contrast, mesenchymal stem cells’ (MSC) proliferative response to PEMF has been chiefly consistent, even with different types of PEMF treatments. In recent years, the effects of electromagnetic force on MSC have been studied several times, and PEMF has been shown to influence the cell cycle, especially the shortening of the lag phase, leading to a higher cellular proliferation index. This may result from different cytokines (e.g., M-CSF, SCF, IL-7) being involved [98]. These proliferative effects of PEMF on MSCs extend for several days after exposure. Importantly, and connecting to the Ca^2+^ ions and ion channels activity with PEMF, it has also been reported that these factors may play a role in the proliferative effects of this treatment modality in stem cells, as they can induce the IGF-1 response [98,99]. Additionally, the expression of FGF, TGF-b, and c-Jun have been shown to be induced by PEMF, presenting other molecular factors involved in the proliferative response to electromagnetic fields [98]. Nevertheless, it is important to point out that, even with a more consistent proliferative PEMF-induced response of MSCs, there is work showing contradicting results, mainly related to the duration of the exposure [98].

##### Effects in Other Cell Types

Additional to MSCs, similar induction of proliferation has been seen in other human cell types modeling normal physiological conditions (e.g., adipose-derived stem cells (MCSs discussed above), tendon stem progenitor cells (TSPCs)) [100,101]. In pathological states such as cancer, PEMF combined with substance treatment (mainly hormones) has been shown to induce cancer cell proliferation [12]. This suggests that PEMF effects might be of higher therapeutic relevance when in combination with other treatments.

#### 3.2.3. Differentiation

Over the past few years, many findings have been published and reviewed with regard to the effects of PEMF on cellular differentiation. As another important process of a cell’s activity and fate, this intricate process is regulated by many different molecular pathways. Here, we will summarize the ones that have been linked to PEMF-induced effects, mainly in stem cells, bone and cartilage, and cancer cells, although other cell types will be mentioned, too. In addition to what has been already mentioned with regards to apoptosis, the changes in intracellular Ca^2+^ levels and other ion dynamics, as well as the effects of adenosine receptors, membrane channels, NO and ROS levels after PEMF treatment, have also been connected to cell differentiation [4,12,19,24,90,93,100,102]. Several reviews have focused on the connection of these effects, linking what is believed to be the initiating effects mentioned above and their relation to cell differentiation [10,24,35,98,103]. Others have published collections of resources that provide significant information on PEMF studies where cell differentiation has been included as a result parameter [4,19,20,33,35,43,93,100,102,104]. As seen in their works, the frequency and dose, as well as the biological state of the given PEMF treatment, span a wide range, which means that, even though some common interpretations can be inferred from the number of studies made, they are not conclusive [33,100].

##### Effects in MSC, Osteoblasts, and Cartilage

Stem cells have been a particularly often-studied cell type in PEMF-related studies, mainly because of their use in tissue regeneration [12]. Especially, MSCs isolated from human bone marrow (hBMSCs) have been used [17]. The studies are inconsistent, as they show both proliferative effects and early-stage differentiation of the cells [35]. Also, they show, once more, that adjuvant elements, such as particular medium conditions, are part of the PEMF-driven response [33,90,100]. Here, it was shown that, if the media is conditioned to push osteogenic differentiation, the result is enhanced with PEMF. The same can be concluded from chondrogenic differentiation studies [93]. This, again, points to the question of whether PEMF-driven effects are only elicited when a certain biological process is already happening. Nevertheless, even with the lack of consistency in experimental setups and the fact that no clear molecular mechanism for PEMF-induced stem cell lineage commitment exists [33,102], researchers have still been able to establish that Runt-related transcription factor 2 (Runx2)/core-binding factor α1 (Cbfa1) and osterix (Sp7) serve as predominant transcription factors committing MSCs to osteogenic differentiation. In contrast, SRY-box transcription 9 (Sox9) and the PEMF-driven modulation of the Wnt/β-catenin signaling pathway are important for chondrogenesis [98]. In other studies, PEMF exposure also triggered a strong expression of osteogenic markers, such as osteonectin, osteopontin, collagen I, and collagen III, suggesting a modulation of the microenvironment, as well as cell differentiation [19,98]. Interestingly, Varani et al. also mentioned a study where the effects of a certain direction of the magnetic field can lead to enhanced MSC chondrogenic differentiation [93]. Also, in the presence of chondrogenic inductive factors in the medium (conditioned media), PEMF has induced collagen type II (Col2) expression, aggrecan, and glycosaminoglycan (GAG) content [21]. Furthermore, PEMF has been shown to increase the expressions of Notch4 and Hey1 during osteogenic differentiation of MSCs, suggesting that the Notch pathway, important in cellular fate and bone development, is activated by PEMF in stem cells [12,48]. Studies (reviewed recently by Varani et al.) have also shown the involvement of MEK/ERK in PEMF-induced osteogenic differentiation MSCs, as well as the activation of p38 MAPK, which is, importantly, connected to the modulation Runx2 [93]. PEMF effects on calcium membrane channels have been associated with the upregulation of gene coding for members of the TGF-beta/BMP superfamily. All these genes can promote the differentiation of MSCs into osteoblasts and the synthesis and bone extracellular matrix [1,44]. Additionally, the same studies mention the connection to adenosine (and its receptors)-driven stem cell differentiation. This link to MSCs differentiation towards both chondrogenesis and osteogenesis, (i.e., A2A and A2BARs activation and A2AARs/CD73 regulation) also hints to a receptor-specific differentiation, where A2A activation would lead to a chondrogenic outcome. Finally, a recent study revealed that certain miRNAs are involved in MSC osteogenic differentiation [105], connecting this important regulatory process once more to PEMF response. To put the many mentioned differentiation effects in perspective, Waldorf et al. [84] reviewed, among others, an interesting study where a microarray analysis of PEMF-stimulated stem cells was conducted. It showed that PEMF clearly affects cells in the mineralization phase the most (cell adhesion and binding proteins), while cells in the differentiation phase are the least affected [106]. Even though this type of PEMF study needs to be repeated with different cell types, it highlights the importance of understanding what these regulatory effects lead to. PEMF’s influence on the mineralization phase of osteogenic differentiation also leaves questions open, as some studies suggest that the treatment increases Ca^2+^ deposition, while others argue that PEMF does not affect this phase in differentiation [100]. A reason for this discrepancy might be the lack of consistency in the studies’ physical and biological setups, and the fact that many studies have used PEMF as an adjuvant to an existing treatment, which can add to the many different outcomes, may also contribute to the discrepancy. Interestingly, PEMF’s trigger of MSC differentiation is not restricted to bone or cartilage, but has also been reported in cardiogenic and neurogenic differentiation [33]. In non-mesodermal lineage cells, the pulse frequency seems to play a pivotal role in the differentiation outcome [98]. Most of the already mentioned molecular pathways for MSC differentiation towards bone and cartilage lineage have also been detected in osteoblastic, osteoclastic, and chondrocytic differentiation. Due to the plethora of important signaling pathways that have been connected to bone and cartilage differentiation after PEMF, it is essential to note that studies have shown that the stimulation of these pathways and molecules can also lead to proliferation. Such is the case of BMP2 effects (part of the TFGb/BMP superfamily), which have been reported to be significantly changed by PEMF, and not only stimulates differentiation of osteoblasts, but also proliferation. Of additional importance is the canonical Wnt pathway, which is known to strengthen osteoblast differentiation and, at the same time, inhibit osteoclast differentiation. BMP and Wnt are also suggested to have synergistic effects on osteoblast differentiation, meaning that, if PEMF can trigger both pathways, the effect might be more potent [48], leaving future studies to understand what such effect might lead to in bone homeostasis. Furthermore, the PEMF effects on the MEK/ERK pathway, seen with MSCs, are conflictive in osteoblast differentiation, as the effects also elicit proliferation and survival. Also, the ERK p38 MAPK-driven differentiation stimulated by electromagnetic field treatment has been connected to elicit both osteoblastic and osteoclastic maturation by different treatment conditions [4], meaning that it is crucial to further study these, and other, multifaceted effects to be able to use PEMF properly in tissue repair. Yuan et al. review adds insulin growth factor (IGF) signaling to the list of PEMF-relevant pathways in osteoclast differentiation where the mRNA expression of IGF-1 is affected [4]. Normally, IGF-1 is required for maintaining the interaction between the osteoblast and osteoclast to support osteoclastogenesis by regulating RANKL and RANK expression [107]. It is, therefore, worth mentioning that, in situations such as tissue damage, inflammation, or diseases (e.g., osteoporosis), this cellular balance can be disrupted, and PEMF could therefore contribute to restore the normal interaction between these two types of cells. In fact, due to the stimulatory effect on osteoblastic differentiation by PEMFs, it has been thought of as a potential candidate for the prevention and treatment of osteoporosis [96].

##### Effects in Other Cell Types

The vast scope of cells studied under PEMF goes well beyond those discussed up to this point. PEMF’s differentiation effects have also been studied in other cells, such as oligodendrocyte precursor cells and PC12 pheochromocytoma cells [10,15,43], where epigenetic changes and the connection to adenosine receptors have been reported [10,96]. Also, the promotion of myotubes and differentiation of skin fibroblasts have been studied [102,108].

### 3.3. Concluding Cellular and Molecular Response Effects of PEMF

All cellular actions described so far are part of almost all tissue and cellular processes in the body and play a vital role in the intricate physiological processes, such as tissue regeneration and inflammatory response, with the addition of other cells, pathways, and signaling processes, which are important for cell recruiting, migration, and communication. Ganesan et al. summarizes the effects of PEMF in different cell types (lymphocytes, neutrophils, osteoblast, osteoclast, chondrocytes, and fibroblast), giving a good overview on the connection of PEMF response to a multidimensional response network [21]. Interestingly, the MSC-based studies have been key to understanding the potential regenerative capacity of PEMF, where results have shown how these cells can modulate and change the course of processes such as inflammation, tissue repair, and establishment of homeostasis [12,48]. We can also appreciate that there are already many studies pointing to the effects of PEMF on bone cells. In the future, these molecular parameters might help identify new tissue repair therapies by more precisely understanding bone phenotypes and disease conditions; for example, choosing the optimal PEMF regime to boost proliferation and differentiation of osteoblasts, while inhibiting osteoclast differentiation and strengthening bone mass [19]. To keep things in perspective, though, the observations made in the work of Mansourian et al. concluded that most of the experiments were carried out on human cells, and, out of 2421 human cell experiments, cell changes were observed in only 51% of the studies. There, they also reaffirmed the differences in PEMF cellular response. Still, even with such inconclusive results, their data provides a starting guideline on the physical and experimental parameters for which to expect a cellular response, and can hopefully assist future groups in their experimental parameter decisions.

## 4. PEMF Effects in the Clinic

Considering the molecular and cellular effects studied in relation to PEMFs with the observed (clinical) results, it is not surprising that this form of treatment has been used in various clinical settings. The link to early clinical trials paved the way for studying the effects of electromagnetic fields on the human body, which have been used primarily in orthopedics and traumatology [14,19,23,25,30,38,47,88]. However, attempts have also been made to treat neurological disorders [10] and wound healing disorders [11,32,109]. This has led to many trial-and-error studies, in which no consistent treatment strategy has emerged.

These differences relate to the physical parameters chosen and the clinical application variables. The clinical variables include therapy frequency (e.g., daily), total therapy duration (e.g., six weeks), and individual treatment duration per session for the different indications. This leads to a very heterogeneous data situation, as shown in a large meta-analysis of randomized and controlled trials [38]. The wide range of therapeutic variables makes it difficult to find a “common” treatment regimen with standardized parameters. While Peng et al. calls for further preclinical and clinical dose-response studies to evaluate the optimal parameters, we believe that multiple clinical trials using the same parameters are needed to estimate efficacy better. Nonetheless, the current study design discrepancy should be an essential consideration in future clinical trials.

In the following, we present a review of clinical trials conducted in recent years to provide a systematic overview of a large number of clinical trials on PEMF therapy. In addition, we also highlight the heterogeneity of the treatment variables applied, all of which are referred to as “PEMF” in the literature. The sections have been divided according to the clinical areas in which PEMF has been used.

### 4.1. Orthopedics

Non-invasive and safe PEMF stimulation is used in the United States and Europe to promote bone regeneration in the clinic. There are two main areas of application; for one, electromagnetic therapy can be used early in acute fractures, namely, when a fracture may be at risk of non-healing due to intrinsic and/or extrinsic factors; for two, an established fracture non-union can also be treated with PEMF.

A systematic review by Hannemann et al. [25] examined randomized controlled trials of the effect of PEMF or pulsed low-intensity ultrasound (LIPUS) on stimulating bone growth in acute fractures compared with placebo. They concluded that PEMF could reduce acute diaphyseal fractures’ radiographic and clinical healing time. However, the pooled data failed to show a significant difference in the proportion of resulting non-unions between groups. They also point out that the results should be viewed critically because of the heterogeneity of the different studies. For example, different fracture types (e.g., tibia, femoral neck, and scaphoid) have specific healing characteristics that make comparison difficult. In addition, the criteria for a positive result were based on different readouts, ranging from a plain radiograph to CT scans and bone density measurements [25]. In the systematic review and meta-analysis on fracture healing in general by Peng et al. [38], the authors concluded that there was only moderate evidence that PEMF increased healing rates and reduced pain. For the former, the risk ratio was 1.22 (95% CI = 1.10–1.35), based on a random-effects model, to increase the overall healing rate with moderate heterogeneity compared with the control group. In a subgroup analysis stratified by fracture age, the delayed and non-healed fractures have a better risk ratio of 1.64 (95% CI = 1.21–2.22) compared with the fresh fractures (<6 weeks) with a risk ratio of 1.20 (95% CI = 1.11–1.29) with the same heterogeneity of the data. Regarding the morphologic classification and method of bone injury, the subgroup analyses showed similar results. Considering time at fracture healing, a risk ratio of −1.01 (95% CI = −2.01–0.00) showed favoritism of the control group compared to fractures treated with PEMF, regardless of subgrouping. They also noted that better parameters for dose and duration need to be determined to better analyze the efficacy of PEMF.

Balvantray et al. [13] reviewed 69 clinical trials of electrical stimulation, including PEMF. In these clinical trials, 73% reported positive outcomes, although they did not differentiate the three different therapies (direct current (DC), capacitive coupling (CC), and PEMF), although most trials (60%) used PEMF. Interestingly, they also examined why more orthopedic surgeons were not using electrical stimulation, including PEMF, in their clinical work, and concluded that their biggest concerns were the inconsistent results of the studies and the cost.

The heterogeneous parameters were very striking in the paper by Daish et al. [19]. Their in vivo (experimental) studies analysis showed that frequencies ranged from 0.1 Hz to 63 kHz, intensities covered 35 µT to 0.03 T, and treatment durations ranged from 15 min to 680 h. This range within the reported parameters made it nearly impossible for the authors to conclude the clinical treatment strategies to be used. However, in the treatment of non-healing tibial fractures, the included prospective studies showed a cure rate between 60% and 88%, with a treatment duration of 3–20 h/day over 8–29 weeks.

In a double-blind, randomized, multicenter study (therapeutic level 1) [110] involving six trauma hospitals, acute tibial shaft fractures (*n* = 259) were treated with electromagnetic stimulation in addition to initial therapy (conventional or surgical). Results were compared with a placebo group. No statistically significant difference was found between the groups when recording the rate of revision surgery within one year (=primary outcome parameter), due to failure of bone healing. Regardless of the therapy, however, compliance with the prescribed treatment of the participants was only moderate and averaged 6.2 h per day, comparable in both groups. However, patients were referred for 10 h per day over a 12-week period, which appeared to be a relatively large expense for patients.

Few studies address this issue when searching for cost analysis in treating acute or non-healing fractures. Hannemann et al. [25] calculated the costs of treating acute scaphoid fractures. No difference in hand function (using the patient-rated hand and wrist evaluation) was found in the clinical outcome. In the economic calculation, the PEMF treatment was significantly more expensive in the average total health care costs per patient (875 € in the placebo group and 1594 € in the active PEMF group).

### 4.2. Osteoarthritis

Osteoarthritis (OA) is a common and debilitating joint disease that affects millions of people globally. Current treatments for OA focus primarily on symptom management, but often do not provide lasting relief.

In a recent review paper with meta-analysis [42], 11 prospective randomized trials involving 614 patients were identified after an informed selection process. Critical OA-associated symptoms, such as pain, stiffness, and physical function, were identified, and the effect of PEMF was determined. Pain indicators, such as WOMAC and VAS, showed a significant reduction compared with the baseline. The parameters of stiffness and physical function in the WOMAC score also showed significant improvement compared with the control intervals.

Not surprisingly, another meta-analysis conducted slightly earlier in 2020 [47] came to a similar conclusion, as 8 of the 15 included studies overlapped with the aforementioned study. Again, the analyses show significant pain relief, without, however, significantly improving stiffness and function, at least in the <4 weeks of PEMF therapy. This meta-analysis also accounted for 3 papers evaluating quality of life, but only one of these showed a clearly significant improvement in quality of life, again, in the <4 weeks of PEMF therapy compared with the control group.

A year earlier, a group from China also conducted a meta-analysis of prospective randomized trials [18], which included 8 studies. It is interesting to note here that this review considered a study from 2002 [111], which was not included in the previously mentioned recent analyses. This, in turn, gives rise to discussion about the inclusion criteria or completeness of such work.

In addition, there is a review of systematic reviews from 2022 from Crevenna’s group [29] that found 10 such papers fitting their inclusion criteria. Of these, half of these reviews show positive results with PEMF in the treatment of osteoarthritis. It is interesting to note here that many papers show a very heterogeneous indication regarding treatment protocols.

A narrative review from Italy in 2021 [34] asked whether PEMF was specifically tested in athletes. However, no such publication could be found in the literature that showed PEMF to be superior as an alternative to sole therapy for osteoarthritis.

Finally, a group, again from Italy [31], addressed the issue of biophysical therapy options for early osteoarthritis and included, among others, laser, ESWT, and PEMF. Including a preclinical and a clinical study, the authors concluded that PEMF has a protective effect on progression (experimental), and, especially, patients <45 years benefit with pain relief and functional improvement, especially in the first year.

### 4.3. Osteopenia

Osteopenia and osteoporosis are diseases characterized by a decrease in bone mineral density, leading to a weakening of the bones and an increased risk of fractures. Osteopenia is considered a precursor to osteoporosis, in which bone density continues to decrease, leading to a higher risk of fracture. Non-invasive treatments play a critical role in the management of both conditions. Treatment of osteopenia and osteoporosis requires a comprehensive approach, with patient education, regular monitoring, and individualized treatment plans, to minimize fracture risk and maintain bone health.

In the experimental approach to this topic, a review paper from 2021 [46] identified a total of 24 viable studies with evaluated parameters from bone mineral density to biochemical analyses, and from histological/histomorhometric workups to CT scans. Of these, 23 showed a positive effect of PEMF on the parameter(s) evaluated in each case.

A review article [45] identifies PEMF as a promising agent for pain relief in osteoporosis patients. Studies show improvement of bone mineral density in different locations, at least in the short-term, but with controversy in the long-term follow-up. Some studies suggest that PEMF treatment may be as effective as alendronate, with certain parameters. However, there is some variability in the results, again attributed to the use of different treatment parameters and small sample sizes. Regardless of bone mineral density, however, PEMF stimulates osteogenesis, as demonstrated by increased biomarkers of serum osteocalcin (OC) and carboxyterminal propeptide of type I collagen.

Another review on electrical stimulation in osteoporosis locates clinical evidence that PEMFs may alleviate osteoporosis-related pain [49], and that bone mass and, thus, osteoporosis could be favorably influenced by PEMF through a RANKL/OPG and Wnt/-catenin pathway. Although the FDA has not yet approved PEMF for the treatment of osteoporosis, based on the current experimental and clinical data, this non-invasive procedure could be an effective adjunct in this indication.

### 4.4. Neurology

Although not as commonly used as in orthopedics, PEMF therapy has also been used in neurology, mostly known as Transcranial Magnetic Stimulation (TMS). Funk et al. [10] made a comprehensive summary of the effects experienced in clinical trials using magnetic fields, including PEMF, for treating neurological diseases. Specifically, for Alzheimer’s disease, pulsed electromagnetic fields have been shown to reduce inflammation and produce vasodilatory effects, improving blood circulation. They also summarized other clinical parameters that improved in neurodegenerative diseases after electromagnetic therapy.

However, based on the current promising clinical studies available, it would be of interest to perform further analyses on pathological neurological conditions of variable etiology.

### 4.5. Wound Healing

PEMF therapy has also been used to treat wounds. Strauch et al. [11] reviewed the use of PEMF in plastic and reconstructive surgery cases. They suggested that PEMF affects pain relief after surgery and reduces swelling. Also, Palmieri et al. [109] made a narrative review on the clinical effects of PEMF on wound healing, where the work of Kwan et al. [112] was highlighted, which showcased the effect of PEMF on chronic diabetic foot ulcers. The volunteers were randomly allocated to the PEMF or the control group. The treatment consisted of 14 sessions over three weeks, with a field frequency of 12 Hz and an intensity of 1.2 mT. After one month, the treated group showed an 18% decrease in wound size, compared to a 10% decrease in the control group.

Even though the studies show good to moderate results, good quality and clinically relevant wound healing studies with PEMF are very sparse. In addition, the existing studies are based on a small sample size, and, once again, the technical parameters of the PEMF device are often missing [11,109,112].

### 4.6. Oncology

The use of PEMF is still very limited in clinical oncology. Vadala et al. [43] reviewed the existing literature on PEMF therapy in clinical oncology, where one study looked at brain tumors, hepatocellular carcinomas, and more. Patients applied themselves PEMF for 60 min with a predefined tumor-specific frequency. Four of the 28 patients presented with stable disease, meaning no progression in tumor growth or new metastasis.

Another study by Costa et al. [113] showcased the use of PEMF on 41 patients with advanced hepatocellular cancer. They received three sessions, with each session lasting 60 min, with frequencies ranging from 100 Hz–21 kHz. Five patients reported a complete disappearance of pain shortly after the treatment. No study patient complained of adverse events associated with the treatment. They conclude that PEMF has a significant anti-tumor effect in the reduction of growth and provides relief of pain in patients with hepatocellular carcinoma.

### 4.7. Concluding Remarks Regarding Clinical Applications

Regardless of the reviews considered and part of the studies, almost every article discusses the heterogeneity of the parameters used. In this context, the sometimes dramatically different selected physical treatment parameters and the treatment variables are an impetus for criticism.

However, positive effects are repeatedly reported if one looks at the clinical results in the various indications, above all in orthopedics and traumatology, in bone healing disorders. From these two findings, it is logical to call for additional studies, in which it would be desirable to use, at least initially, the same parameters to ensure better comparability. Subsequently, it would make sense to systematically vary the various parameters individually until the best possible success is achieved with this setting.

## 5. Future Perspectives

PEMF-generated magnetic fields can penetrate all tissues with little to no attenuation. They may cause biological responses, leading to molecular and physiological changes in many parts of the human body. Physical parameters and definitions need to be settled to understand the full scope of those aspects of PEMF. When this critical part is clear, the focus should turn to understand cellular and molecular mechanisms at the tissue level. That means using in vitro modeling techniques, which allow a connective and inclusive view of the cellular and molecular processes involved in PEMF response. These methods could include any 3D in vitro modeling approach already used for different organs, including microfluidic systems. This is important, as the effects of PEMF on intercellular signals at the microvesicle and feedback loop level could give relevant insights into how this type of therapy can be used. This is not to forget the importance of the micro-environmentally driven impact on the cellular response, especially in tissue repair and inflammation. This may be particularly important, as the induced cellular and tissue interactions with the pulsed magnetic field occur on many length scales and corresponding frequencies. Additionally, the modeling of healthy tissue, and pathological or disease state modeling should lead to understanding not just the medical use of PEMF, but also its restrictions. For example, treatment in malignancies might show ways to use PEMF as an additional therapy accompanying a pharmacological approach. Also, AI, being implemented in bioinformatics, could lead the way to use mathematical-simulating tools. This would enable us to see how the cellular and molecular effects of PEMF can lead to specific network patterns. These patterns can then provide new treatment strategies in bone and cartilage repair, and in other organs of interest. Daish et al. and Thielscher et al. have summarized the work done so far on this front, where mathematical models of tissue differentiation and vascularization have been developed [19,114].

Other potential target groups have also been discussed in orthopedics, where PEMF and related techniques are already used, irrespective of the clinical indication. One such group can be identified as the increasingly older population, in which age-associated diseases and injuries can be expected to increase. In addition to osteoporosis in older generations, the rise in comorbidities such as obesity, diabetes, and degenerative joint disease would lead to increased orthopedic interventions in osteosynthesis and joint replacement implants. The global orthopedic implant market has been predicted to grow at an annual rate of about 5.35% from 2022 to 2029. This also implies that an increase in the absolute number of implant loosening can be expected. Based on this projection, methods are needed to treat such loosening effectively, or, even better, to prevent them from happening. Biophysical therapies, including PEMF, have a great potential to promote early osseointegration. Moreover, these low-cost, non-invasive applications have a favorable risk-benefit profile. One such application is the first clinical result of oral implants where these therapeutic methods have succeeded [16,26,115]. For orthopedic applications, however, more studies are needed on the interaction of therapeutic magnetic fields with various implants to make a better interpretation of these therapies’ effects. Due to the current use of medical titanium alloys in implants, the magnetic response is weak—especially when compared to the ferromagnetic implant materials that were clinically prevalent in the past. Although no relevant magnetization or temperature increase is expected with the new titanium alloys, it is still subject to the Lenz effect. Before routine application, studies must clarify which interactions between the electromagnetic fields and the implant occur, and what effects these have, both locally and, possibly, systematically. Based on such studies, adequate clinical parameters must be defined, and recommendations for appropriate indications must be made.

Even more relevant may be the prophylactic use of PEMF; for example, when it is adjuvantly applied during initial osteosynthesis in fractures with a high risk of developing non-unions (e.g., open tibia fractures). The avoidance of complicated fracture healing in patients has enormous savings potential for health insurance funds and increases the quality of life of the affected patients. The same line of reasoning can easily be extrapolated to joint replacement. Similarly, implants and bone substitutes may also be favorably influenced by PEMF, especially in cases of compromised bone quality or restricted local blood supply, as experimental studies have already shown. Interestingly, in osteoporosis itself, some PEMF studies already showed significantly increased bone mineral density when combined with conventional drugs [116], which would potentially change the way postmenopausal osteoporosis is treated.

In conclusion, even though the lack of consistent study parameters makes PEMF effects scientifically challenging to evaluate, this non-invasive and comparatively inexpensive treatment tool, which does not require any additional infrastructure, has been shown to positively contribute to difficult clinical conditions in orthopedics and traumatology. In particular, effective adjuvant options to the respective standard therapies in orthopedics and other medical fields stand to change future care strategies and patient outcomes.

## Figures and Tables

**Table 1 ijms-24-11239-t001:** Overview of review papers on PEMF referenced in this paper. Indicated number of references are those with indexed DOIs in Crossref.

Publications	Year	References (With Indexed DOIs)	Focus
Barati et al. [12].	2021	212 (204)	Apoptosis
Bhavsar et al. [13]	2019	171 (147)	Bone healing
Cadossi et al. [1]	2020	62 (60)	Mechanics of PEMF in bone healing
Caliogna et al. [14]	2021	54 (0)	Bone healing
Capone et al. [15]	2021	47 (45)	Neuroprotective after ischemic damage
Cecoro et al. [16]	2022	74 (65)	Dental implant osseointegration
Chalidis et al. [17]	2011	27 (24)	Mechanisms of PEMF in bone fractures
Chen et al. [18]	2019	36 (0)	Osteoarthritis
Daish et al. [19]	2017	113 (102)	Bone healing
Di Bartolomeo et al. [20]	2022	98 (93)	Mechanics of PEMF in bone healing
Funk et al. [10]	2021	83 (0)	Magnetic fields on neurological diseases
Ganesan et al. [21]	2009	82 (0)	Pain management and improvement in arthritis
Giorgi et al. [22]	2021	119 (116)	Epigenetic alternation caused by magnetic fields
Gossling et al. [23]	1992	79 (47)	Bone healing
Gualdi et al. [24]	2021	122 (111)	Mechanisms in wound healing
Hannemann et al. [25]	2014	33 (32)	Acute fracture healing
Khan et al. [26]	2022	29 (0)	Dental implant osseointegration
Mansourian et al. [27]	2021	101 (90)	PEMF effect on cells
Markov et al. [28]	2006	173 (166)	Mechanism for pain control
Markovic et al. [29]	2022	27 (24)	Osteoarthritis
Massari et al. [30]	2019	74 (65)	Bone and cartilage
Mauro et al. [31]	2021	26 (26)	Osteoarthritis
Mayrovitz et al. [32]	2022	72 (70)	Magnetic fields in diabetic complications
Maziarz et al. [33]	2016	49 (44)	Electromagnetic fields on stem cells
Moretti et al. [34]	2021	76 (2)	Joint degeneration
Pagani et al. [35]	2017	85 (0)	Complex regional pain syndrome
Panagopoulos et al. [36]	2002	61 (49)	Mechanism of electromagnetic fields
Peng et al. [37]	2021	37 (33)	Angiogenesis
Peng et al. [38]	2020	57 (42)	Bone healing
Rahbek et al. [39]	2004	24 (19)	Physical mechanism and tissue interaction
Rossi et al. [40]	2009	297 (279)	Safety of TMS
Shupak et al. [41]	2003	115 (15)	PEMF in clinics
Strauch et al. [11]	2009	63 (0)	Pain and edema
Tong et al. [42]	2022	42 (40)	Osteoarthritis
Vadalà et al. [43]	2016	105 (90)	PEMF in oncology
Vicenti et al. [44]	2020	49 (45)	Bone healing
Wang et al. [45]	2019	101 (0)	Osteoporosis
Wang et al. [46]	2021	78 (73)	Osteopenia
Yang et al. [47]	2020	56 (48)	PEMF in osteoarthritis
Zhang et al. [48]	2020	89 (87)	Electromagnetic fields on bone cells
Zhang et al. [49]	2023	78 (0)	Osteoporosis

**Table 2 ijms-24-11239-t002:** Comparison of minimal invasive therapy methods relying on different physical principles.

Therapy	Physical Field	Note	Scope of Application	Limitations	Indication
PEMF (Pulsed Electromagnetic Field)	Electromagnetic Field	A coil is used to rapidly create an electromagnetic field	No contraindicated tissue	Can heat up metallic implants and thermally destroy adjacent tissue; avoid application close to pacemakers [40]	Wound healing disorders, Non-unions, Pain management [41]
HI-PEMF (High Intensity Pulsed Electromagnetic Field)	Electromagnetic Field	PEMF with much higher energies	No contraindicated tissue	Can heat up metallic implants and thermally destroy adjacent tissue; avoid application close to pacemakers [40]	Fractures, nerve injuries, pain reduction [59]
TMS (Transcranial Magnetic Stimulation)	Electromagnetic Field	PEMF in the use of treating the brain	Applied on brain tissue	Additional to limitations of PEMF, short-term nausea and vertigo possible, but no permanent harmful effects; uncertain risk of induced epilepsy; avoid at pregnancy [40]	Alzheimer’s, depression, pain management [60]
TENS (Transcutaneous Electrical Nerve Stimulation)	Electric Field	Electrodes placed on skin create an electrical field, exciting nerves	Nerve stimulation, can result in muscle movement	Difficulty in penetrating into deeper tissue [39]; avoid at pregnancy, epilepsy, and close to pacemakers [61]	Muscle stimulation, pain relief [62]
US (Ultrasound)	Mechanical Energy	Continuous ultrasound signal as tissue stimulating therapy, in the MHz frequency range	Most tissue types, keep away from eyes and lungs, as pressure reflection may damage tissue	Tissue interfaces may oppose a barrier, can heat up tissue if statically applied	Tendinopathy [63], bone repair, [64]
ESWT (Extracorporeal Shockwave)	Mechanical Energy	Single pressure shock wave—higher Frequencies, as in ultrasound	Most tissue types, keep away from eyes and lungs, as pressure reflection may damage tissue	Tissue interfaces may oppose a barrier	Tendinopathy [63], erectile dysfunction [65], pseudoarthrosis [66]
LIPUS (Low Intensity Pulsed Ultrasound)	Mechanical Energy	Pulsed ultrasound in the lower MHz frequency region, with low intensity	Most tissue types, keep away from eyes and lungs, as pressure reflection may damage tissue	Tissue interfaces may oppose a barrier	Fracture healing [67], bone nonunion, soft tissue regeneration [68]
HIFU (High Intensity Focused Ultrasound)	Mechanical Energy	High energy ultrasound in the higher MHz frequency region, thermally ablates target [69]	Most tissue types, keep away from eyes and lungs, as pressure reflection may damage tissue	Tissue interfaces may oppose a barrier	Minimal invasive surgery alternative [64]

**Table 3 ijms-24-11239-t003:** Overview of basic necessary parameters to quantify applied magnetic fields.

	Symbol	Unit	Description
Magnetic flux density	*B*	Tesla	Intensity of the magnetic field
Pulse duration	Δt	s	Duration of a single pulse or pulse train
Pulse repetition frequency	f	Hz	How often a pulse or pulse train is repeated per second
Field frequency	$f_{f}$	Hz	Main frequency of the signal, can be estimated using the zero-crossings, or, better, by calculating the frequency spectrum of a single pulse or pulse train
Pulse gradient	δB/δt	T/s	Maximum slope of the magnetic pulse. It correlates with the induced electrical current in the tissue
Pulse signal (Plot)	-	-	Plot of the pulse signal, includes all of the mentioned parameters, and can be used as a replacement

**Table 4 ijms-24-11239-t004:** A representative selection of clinical devices, illustrating the wide range of generated magnetic fields, although, for most, essential parameters are missing, i.e., not readily available.

Device	Maximum Pulse Magnitude *B* [T]	Pulse Repetition Frequency f [Hz]	Pulse Duration Δt [ms] [77]	Field Frequency [Hz]	Pulse Shape	Pulse Gradient δB/δt [T/s]
Zimmer—emFieldPro [78]	3 T	1–150 Hz	variable	variable	variable	-
PEMF-120 [79]	0.94 T	1–50 Hz	-	-	-	-
Magnetolith —Storz [80]	0.2 T	≤10 Hz	-	100–300 kHz	Dampened sine wave	65,300 T/s
Hofmag [81]	0.029 T	6 Hz	1 ms	28 kHz	Dampened sine wave	-
Biostim—IGEA Medical [20,82]	0.002 T	75 Hz	1.3 ms	-	Trapezoidal-shaped signal	-
BIOMET—EBI Medical Systems [20]	0.0016 T	15 Hz	5 ms	-	Trapezoidal-shaped signal, pulse train	-
PST [83]	0.0015 T	10–20 Hz	-	-	quasi-rectangular	-
SpinalStim— Orthofix Inc [84]	0.00068 T	1.5 Hz	-	3.85 kHz	Triangle-shaped signal, pulse train	-
SofPulse [52,85]	0.000005 T	5 Hz	2 ms	27 MHz	Undefined signal, burst mode	-

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
