# Peer review of "Pulsed Electromagnetic Fields (PEMF)—Physiological Response and Its Potential in Trauma Treatment"

_ijms, 2023, doi:10.3390/ijms241411239_

Round 1
Reviewer 1 Report
The manuscript# ijms-2402027 entitled “PEMF - physiological response and its potential in trauma 2 treatment” by Flatscher J et al and colleagues is a timely review of the exciting field. The use of biophysical agents as therapeutic modalities is increasing in popularity but has also brought more rigorous attention to the methodology and inconsistencies in biological responses that are well highlighted by the authors in this work. There are a few issues that could be addressed to ensure an appropriate interpretation of this work.
1. Please use the full term and not the acronym in the title
2. The introduction is well written. However, the authors could add a few statements to describe if isolated magnetic fields and electrical field treatments differ from PEMF. It remains unclear how PEMF is different from ESWT, LIPUS/US, TENS, and TMS. These definitions would be helpful, perhaps as an easy-reference table/image.
3. A tabular outline of definition, units and explanation of the duty cycle, pulse frequency, and field/carrier frequency would be helpful.
4. Figures are completely missing. An outline of the pulse shape, common PEMF devices, and clinical use would be helpful.
5. The mechanistic sections could be further refined as it currently reads very inconclusively. Using the loss of function evidence, can the authors summarize the evidence for adenosine (not Adenine Line 264), calcium signaling, and Redox
6. Several minor typos and details are missing in various sections that could benefit the readers. For examples
A. What is the magnitude of the mechanically-induced endogenous bone piezoelectric field?
B. What are the PEMF parameters of several studies for apoptosis, cell proliferation, and differentiation?
C. Line 511-512 Hemostatis or Homeostatsis?
D. Future Perspectives has some redundant info on osteogenic effects that are better placed in the prior (4) section.
E. Please comment on the availability of clinical PEMF units and any standardization necessary
7. Overall, sections 3 and 4 could be better streamlined with individual sections being sub-sectioned into specific themes. For eg. 3.2.1 Could address normal versus pre/malignant cells, 3.2.2 could address effects on osteoblasts, and MSCs, 3.2.3. could address bone, cartilage, muscle, neuro, and skin more succinctly.
Good
Reviewer 2 Report
Dear Author,
Review entitled PEMF - physiological response and its potential in trauma treatment is of choice of readers in scientific community however Following points needs to be consider for possible consideration
1. Literature study should be extensive. Cite appropriate number of references specially recent reports in this field.
2. Technology/ Mechanism diagramatic representation is highly appropriate.
3. Diagram/ Tabular data representation is expected.
4. Limitations/Adverse events are expected
5. Section 4. PEMF effects in the clinic: Under each sub section in depth literature and evidences need to include.
Minor editing of English language required
Reviewer 3 Report
Please, see the review and make minor adjustments.

Round 2
Reviewer 2 Report
Dear Author,
1. Literature study should be extensive. Cite appropriate number of references specially recent reports in this field.
We agree that a literature study should be extensive, but our aim was to focus on reviews and summaries to provide researchers with a scaffolded structure of the available literature. We also added citations and an overview of the review papers we cited and their respective focus and statistical data.
Reviewer comment on revised version: Partially Accepted
2. Technology/ Mechanism diagramatic representation is highly appropriate.
We had early on decided against adding figures to the review, as we are trying to give an overview of the technology, as well as describing the mechanism behind it and the clinical usage, adding figures only sporadically would unnecessarily put the focus on this specific paragraph. Adding figures to all the sections (e.g. differences in signal, describing each mechanism as a figure, clinical application) would exceed the scope we set for this review.
Reviewer comment on revised version: Partially Accepted
3. Diagram/ Tabular data representation is expected.
We added tables showing which reviews we observed in more detail, we addeda comparison to other biophysical therapy methods and an overview of PEMF devices used in clinics and in research to show their large variability.
Reviewer comment on revised version: Accepted
4. Limitations/Adverse events are expected
Common important adverse effects are discussed in section 2.6. As electromagnetic fields are widely used and present in daily life in a variety of frequencies and strength, but also used in other medical devices (e.g. MRI), they are considered as save with limited direct harmful effects. In addition, we added the limitations of the some biophysical therapies for a comparison in table 2.
Reviewer comment on revised version: Partially Accepted
5. Section 4. PEMF effects in the clinic: Under each sub section in depth literature and evidences need to include.
We added a better overview of the papers we reviewed in the introduction. As we do not see this publication as a comparison of clinical outcomes, but are more comprehensive overview of what PEMF is, to redirect interested researchers and clinicians if further interests arise and what to look out for when doing research using these device, we do not see the need to compare clinical results further. We did not intend to review the efficacy of PEMF in clinics, as we believe that with the lack of parameters in most publications a comparison is not conclusive, or already has been tried to achieve in one of the mentioned reviews.
Reviewer comment on revised version: Partially Accepted
Minor editing of English language required
Author Response
"Please see the attachment."
